# Efficacy Assessment of Five Policosanol Brands and Damage to Vital Organs in Hyperlipidemic Zebrafish by Six-Week Supplementation: Highlighting the Toxicity of Red Yeast Rice and Safety of Cuban Policosanol (Raydel^®^)

**DOI:** 10.3390/ph17060714

**Published:** 2024-05-31

**Authors:** Kyung-Hyun Cho, Ashutosh Bahuguna, Ji-Eun Kim, Sang Hyuk Lee

**Affiliations:** Raydel Research Institute, Medical Innovation Complex, Daegu 41061, Republic of Korea

**Keywords:** high-density lipoproteins (HDL), hyperlipidemia, fatty liver, inflammation, red yeast rice, monacolin K, octacosanol, policosanol

## Abstract

Policosanol is a mixture of long-chain aliphatic alcohols (LCAAs) derived from various plant and insect origins that are marketed by various companies with distinct formulations and brand names. Policosanols offer several beneficial effects to treat dyslipidemia and hypertension; however, a comprehensive functionality comparison of various policosanol brands has yet to be thoroughly explored. In the present study five distinct policosanol brands from different origins and countries, Raydel-policosanol, Australia (PCO1), Solgar-policosanol, USA (PCO2), NutrioneLife-monacosanol, South Korea (PCO3), Mothernest-policosanol, Australia (PCO4), and Peter & John-policosanol, New Zealand (PCO5) were compared via dietary supplementation (1% in diet, final *wt*/*wt*) to zebrafish for six weeks to investigate their impact on survivability, blood lipid profile, and functionality of vital organs under the influence of a high-cholesterol diet (HCD, final 4%, *wt*/*wt*). The results revealed that policosanol brands (PCO1–PCO5) had a substantial preventive effect against HCD-induced zebrafish body weight elevation and hyperlipidemia by alleviating total cholesterol (TC) and triglycerides (TG) in blood. Other than PCO3, all the brands significantly reduced the HCD’s elevated low-density lipoprotein cholesterol (LDL-C). On the contrary, only PCO1 displayed a significant elevation in high-density lipoprotein cholesterol (HDL-C) level against the consumption of HCD. The divergent effect of PCO1–PCO5 against HCD-induced hepatic damage biomarkers, aspartate aminotransferase (AST) and alanine aminotransferase (ALT), was observed. PCO1, PCO2, and PCO4 efficiently curtailed the AST and ALT levels; however, PCO3 and PCO5 potentially aggravated the HCD’s elevated plasma AST and ALT levels. Consistently, the hepatic histology outcome revealed the least effectiveness of PCO3 and PCO5 against HCD-induced liver damage. On the contrary, PCO1 exhibited a substantial hepatoprotective role by curtailing HCD-induced fatty liver changes, cellular senescent, reactive oxygen species (ROS), and interleukin-6 (IL-6) production. Likewise, the histological outcome from the kidney, testis, and ovary revealed the significant curative effect of PCO1 against the HCD-induced adverse effects. PCO2–PCO5 showed diverse and unequal results, with the least effective being PCO3, followed by PCO5 towards HCD-induced kidney, testis, and ovary damage. The multivariate interpretation based on principal component analysis (PCA) and hierarchical cluster analysis (HCA) validated the superiority of PCO1 over other policosanol brands against the clinical manifestation associated with HCD. Conclusively, different brands displayed distinct impacts against HCD-induced adverse effects, signifying the importance of policosanol formulation and the presence of aliphatic alcohols on the functionality of policosanol products.

## 1. Introduction

Dyslipidemia refers to abnormal blood lipid and lipoprotein levels owing to an impaired lipoprotein metabolism and is typically characterized by elevated total cholesterol (TC), triglycerides (TG), low-density lipoprotein cholesterol (LDL-C), and diminished high-density lipoprotein cholesterol (HDL-C) levels in blood [1]. A sedentary lifestyle and lack of exercise, along with a variety of medications and ailments, are the primary factors leading to dyslipidemia [1,2]. Around 50–60% of people with obesity have been identified with disturbed lipid profiles signifying a deep association of obesity with dyslipidemia [3,4]. Furthermore, dyslipidemia can be attributed to genetic defects of the lipid metabolism [1,2]. The pathophysiology/pathogenesis of several diseases often allied with dyslipidemia, precisely, an imperative influence of dyslipidemia on coronary heart disease and the subsequent progression of atherosclerosis have been firmly established [1,5]. 

Statins are the class of drugs that are often used to cure hypercholesterolemia via inhibition of cholesterol synthesis [6,7]. Despite the several benefits of statins, its adverse effects also have been noticed in a non-negligible percentage in the recipients and pose a safety concern with liver toxicity and muscle injury due to the depletion of coenzyme Q_10_ (CoQ_10_) [6,8,9,10]. To overcome the disadvantage of statins, varied nutraceuticals, and functional foods, such as phytosterols, phytoestrogens, policosanol, red yeast rice, and monacolins, have been recognized for their considerable potential in reducing lipid levels [11]. Among the different nutraceuticals and functional foods, policosanol has extensively been explored as an agent to treat dyslipidemia and hypertension by testing its efficacy in varied preclinical and clinical studies [12,13,14]. Moreover, policosanol in combination with classical statins displayed a substantial effect to minimize the toxic effect of statins [15,16,17] its possible use as a combinational therapy was proposed. Furthermore, a recent study found that a combination of high-intensity exercise and Cuban policosanol intake in obese individuals improved hypertension and dyslipidemia. This improvement was attributed to an enhanced HDL quality and antioxidant functionality, achieved without disrupting the CoQ_10_ metabolism or causing liver damage [18].

The established role of policosanol as platelet anti-aggregation, amelioration of neurological function in ischemic stroke patients, and blood pressure maintenance has been reported elsewhere [14]. Even more, some studies have verified the curative effect of policosanol against neurological disorders like Parkinson’s and Alzheimer’s disease [14]. Nevertheless, with a diverse functionality, Cuban policosanol’s role as a lipid-lowering agent and blood pressure-lowering agent by alleviating blood TC, TG, and LDL-C and raising HDL-C [14,19,20] level has been claimed via cholesteryl ester transfer protein (CETP) inhibition and the improvement of HDL functionality such as its antioxidant activity and cholesterol efflux (CE) ability [21,22,23].

Policosanol is a mixture of long-chain aliphatic alcohol (LCAA) derived from a variety of plant sources such as sugarcane, rice bran, grapes, and wheat germ [14,24]. For the first time in 1993, policosanol was extracted from sugarcane in Cuba, which typically contains eight LCAAs: tetratriacontanol (C34), dotriacontanol (C32), triacontanol (C30), nonacosanol (C29), octacosanol (C28), heptacosanol (C27), hexacosanol (C26), and tetracosanol (C24) [18,25,26]. However, the composition of LCAAs in policosanol varied greatly based on the difference in source material, the source of origin, harvesting time, and the method of extraction [14,27], consequently having a distinct functionality and inconsistency between in vitro and in vivo results [28,29]. Many policosanol products, approximately 32 brands with daily dosages ranging from 5 to 50 mg, have been marked globally as functional foods. These products often claim to treat hypercholesterolemia and dyslipidemia. However, the different sources and brands of policosanol exhibit unequal ingredient compositions of LCAAs, such as the octacosanol content, which may display different in vitro characterizations and in vivo efficacies. Indeed, there has been no in vivo study to compare direct efficacy and toxicity among the various policosanol products available in global sales via supplementation of animal experiments. Because the diverse ingredients of policosanol (major) and additives (minor), such as natto, red yeast rice, or CoQ_10_, it is necessary to conduct a comparison study to evaluate the various policosanol brands.

To address this, the present study aimed to assess the comparative effect of five distinct commercially available policosanol brands on the survivability, body weight, and blood lipid profile of hyperlipidemic zebrafish during six weeks of supplementation. Additionally, the comparative effect of the different policosanol brands was examined on the functionality and damage of the major organs, liver, kidneys, testes, and ovaries in the hyperlipidemic zebrafish.

## 2. Results

### 2.1. Survivability and Body Weight of Zebrafish

The survivability of zebrafish across all the groups during six weeks is depicted in Figure 1A,B. The survivability of zebrafish varied from 67.8% to 96.4% at week 6 of consumption of different diets. The maximum survivability (96.4%) was observed in ND, HCD and PCO1-supplemented groups. On the contrary, lower survivability was observed in PCO2- (89.3%), PCO4- (85.7%), and PCO5-supplemented (78.6%) groups. The most adverse effect on the zebrafish survivability was observed in the PCO3 group, where the survivability started to decline after 7 days of feeding (96%), further reached 82.1% at day 20 and finally reached 67.8% at 42 days (six weeks) of consumption (Figure 1A,B). When compared to ND, HCD, and PCO1-supplemented groups, a significant 1.4-fold (*p* < 0.01) lower survivability was noticed in the PCO3-supplemented group at six weeks of consumption, signifying PCO3’s adverse effect on the survivability of zebrafish.

The body weight analysis revealed a substantial time-dependent enhancement of body weight in the ND and HCD groups during the six weeks of consumption (Figure 1C,D). Contrary to the HCD and ND groups, in all the PCO-consuming groups, body weight either enhanced slightly or decreased with time compared to the initial day’s body weight. The most noticeable effect was observed in the PCO3 group, where the body weight of zebrafish started to decline just after two weeks of consumption and progressively decreased with time and reached 333.7 g at six weeks, which was significantly (34.2%, *p* < 0.001) lower than the body weight observed at the initial day (507.2 g) (Figure 1C). When compared to the HCD group, a significant (*p* < 0.001) reduction in body weight was observed in all the PCO-consumed groups at week 6 (Figure 1D). 

### 2.2. Blood Lipid Analysis

The impact of HCD feeding and the consumption of policosanols on the blood lipid profile is documented in Figure 2. A 53.7% higher TC level was detected in the HCD-consumed group compared to the ND control group. The consumption of PCO1, PCO2, PCO3, PCO4, and PCO5 led to significantly reduced TC levels (*p* < 0.001) at 41.1%, 25%, 34.8%, 24.1%, 44.3%, and 39.4%, respectively, compared to the HCD-only group. Besides TC, a significant effect of HCD consumption on the elevation of the TG level was detected, which was found to be 98.5% higher than the basal TG level detected in the ND control group (Figure 2). The consumption of different PCOs significantly reduced the elevated TG level in response to HCD consumption. The most effective results, with 52.2% (*p* < 0.001) and 58.19% (*p* < 0.001) reduced TG levels compared to the HCD group, were observed in the PCO1- and PCO2-consumed groups. 

A significant decline in HDL-C (15.2%, *p* < 0.01) was detected in the HCD-consumed group compared to the ND group, signifying the adverse effect of HCD on the HDL-C level (Figure 2). The consumption of PCO1 was found effective in improving the HDL-C level impaired by the HCD consumption. A significant (25.2%, *p* < 0.01) enhancement in the HDL-C level was observed in the PCO1-consumed group compared to the HDL-C level in the HCD-consumed group. In contrast, the other PCOs did not impact the elevation of HDL-C levels disrupted by the consumption of HCD. Even more, the consumption of PCO2, PCO3, and PCO4 displayed a significant decline of 15.7% (*p* < 0.05), 45.2% (*p* < 0.001), and 14.5% (*p* < 0.05) in the HDL-C level as compared to the HCD group. Contrary to HDL-C, a significant 2.9-fold (*p* < 0.001) enhancement in LDL-C level was detected in the HCD group compared to the LDL-C level of the ND group, signifying the impact of HCD on the elevation of the LDL-C level (Figure 2). The HCD’s elevated LDL-C level was significantly reduced by 5.1-fold (*p* < 0.001), 1.7-fold (*p* < 0.01), 3.2-fold (*p* < 0.001), and 2.4-fold (*p* < 0.001) with the consumption of PCO1, PCO2, PCO4, and PCO5, respectively. However, the consumption of PCO3 displayed no desirable effect against elevated LDL-C levels from the HCD consumption.

### 2.3. Plasma Hepatic Function Biomarker Assessment

The blood AST and ALT levels were determined to evaluate the effect of the consumption of different PCOs on liver functionality under the influence of an HCD. As depicted in Figure 3, a significant 2.2-fold (*p* < 0.001) elevated AST level was quantified in the HCD-consumed group compared to the ND control group. The consumption of PCO1, PCO2, and PCO4 significantly impacted the HCD’s elevated plasma AST levels. A significant 1.9-fold (*p* < 0.001), 1.3-fold (*p* < 0.01), and 1.3-fold (*p* < 0.01) reduced AST level was quantified in the PCO1, PCO2, and PCO4 groups, respectively, contrasting with the AST level of the HCD group. When compared to PCO2 and PCO4, PCO1 displayed a ~1.5-fold lower AST level, signifying the higher potency of PCO1 over PCO2 and PCO4. Unlike PCO1, PCO2, and PCO4, the consumption of PCO3 and PCO5 augmented the HCD-induced AST level. 

Consistent with the AST findings, a significant 2.1-fold (*p* < 0.01) higher ALT level was detected in the HCD-consumed group than in the ND control group (Figure 3). The consumption of PCO1, PCO2, and PCO4 had the substantial effect of reducing the HCD’s elevated ALT levels manifested by 2.2-fold (*p* < 0.01), 1.7-fold (*p* < 0.01), and 1.8-fold (*p* < 0.01) reduced AST levels, respectively, as compared to the HCD group. On the contrary, a significant 1.6-fold (*p* < 0.01) and 1.5-fold (*p* < 0.01) heightened ALT level was quantified in the PCO3- and PCO4-supplemented groups than the HCD control group.

### 2.4. Evaluation of the Hepatic Section

The outcome of the H&E staining revealed the hepatic degeneration and neutrophil infiltration in the hepatic tissue of the HCD group, which accounted for 26.5% of the H&E-stained area that was significantly (2.3-fold, *p* < 0.001) larger than the H&E-stained area observed in the ND control group (Figure 4A,B,F). The HCD-induced hepatic damage was significantly prevented by the supplementation of PCO1 and PCO2, evidenced by the 11.3% and 14.7% H&E-stained area, i.e., significantly smaller by 2.3-fold (*p* < 0.001) and 1.8-fold (*p* < 0.001) than the H&E-stained area of the HCD group. On the other hand, PCO3, PCO4, and PCO5 did not have much effect on HCD-induced hepatic damage.

The ORO staining revealed fatty liver changes in the HCD-consumed group that were significantly (3.2-fold, *p* < 0.001) larger than the ORO-stained area observed in the ND control group (Figure 4C,D,G). On the contrary, all the PCOs significantly (*p* < 0.001) prevented the HCD-induced fatty liver changes, evidenced by a significant ~2-fold (*p* < 0.001) reduced ORO-stained area compared to the only HCD group.

A higher prevalence of senescent cells determined by a senescent-associated β galactosidase (SA-β-gal) assay was observed in the HCD-consumed group, which was significantly (48-fold, *p* < 0.001) higher than the senescent cells observed in the ND-supplemented group (Figure 4E,H). HCD-induced cellular senescence was prevented considerably by PCO treatment. The most noticeable effect was exerted by PCO1, PCO2, and PCO4, manifested by a significant 48-fold (*p* < 0.001), 13.2-fold (*p* < 0.001), and 33.1-fold (*p* < 0.001) reduced senescent stained area compared to the HCD group. PCO3 also documented a significant effect on cellular senescence; however, PCO5 was ineffective in preventing HCD-induced hepatocyte cellular senescence. The results signify that distinct PCOs have a diverse impact on the HCD-impaired hepatic tissue; precisely, PCO1 displayed a substantial beneficial effect against the HCD-induced hepatic alteration of zebrafish.

### 2.5. Assessment of Oxidative Stress, Apoptosis, and Inflammation in the Liver

Reactive oxygen species (ROS) level in the hepatic tissue was examined by DHE fluorescent staining (Figure 5A,E). A massive production of ROS was observed in the HCD group, which was significantly (6.6-fold, *p* < 0.001) higher than the ROS level observed in the ND group. The HCD-induced ROS level was countered considerably by the consumption of PCO1 and PCO4. The PCO1 and PCO4 groups displayed significant 8.4-fold (*p* < 0.001) and 2.2-fold (*p* < 0.01) diminished ROS levels as compared to the ROS level detected in the HCD-consumed group. On the contrary, PCO2, PCO3, and PCO5 displayed no effect against HCD-induced ROS levels. 

The cellular apoptosis extent in the liver was detected by AO-fluorescent staining, which revealed a significant 6.0-fold (*p* < 0.001) higher apoptosis extent in the HCD-consumed group than that in the ND control group (Figure 5B,F). The PCO1 and PCO4 groups effectively inhibited the HCD-induced apoptosis as evidenced by significant 8.1-fold (*p* < 0.001) and 1.6-fold (*p* < 0.05) reduced AO fluorescent intensities compared to the HCD-consumed group. In contrast, PCO2, PCO3, and PCO5 did not improve the HCD-induced apoptosis in the liver.

As depicted in Figure 5D,G, a 10.1-fold (*p* < 0.001) higher production of IL-6 was quantified in the HCD group compared to the ND control group, suggesting the influence of the HCD on hepatic inflammation. In PCO1, a significant 7.2-fold (*p* < 0.001) reduced IL-6 production was quantified compared to the IL-6 level detected in the HCD group, suggesting the effective role of PCO1 in inhibiting HCD-induced IL-6 production. In contrast to PCO1, the other PCOs (PCO2–PCO5) displayed a non-significant effect on the reduction in IL-6 level elevated by the consumption of the HCD. These results documented the efficient effect of the PCO1 consumption in maintaining the HCD-disturbed hepatic inflammation, ROS production, and apoptosis in the liver.

### 2.6. Examination of the Kidney Section

The H&E staining, as depicted in Figure 6A, represents the kidney morphology among the different groups. A well-differentiated and compact proximal and distal tubular structure was observed in the ND control group. In contrast, a disorganized sprawling proximal and distal tubular structure with visible luminal derbies in the tubular cast was observed in the HCD-consumed group, indicating the adverse effects posed by the HCD consumption on the kidney. The consumption of PCO1 and PCO2 was effective against HCD-induced kidney impairment, which is apparent in the compact proximal and distal tubular structure; however, the occasional presence of luminal debris was also noticed. On the contrary, the PCO3, PCO4, and PCO5 displayed no impact on the renal structure, as documented by disorganized and sparsely populated proximal and distal tubules with frequent luminal cell derbies.

A higher DHE fluorescent intensity corresponding to the ROS production was observed in the HCD-consumed group, which was significantly (9.3-fold, *p* < 0.001) higher than the DHE fluorescent intensity observed in the ND control group (Figure 6B,E). The consumption of PCO1, PCO2, PCO4, and PCO5 was found effective and reduced the DHE fluorescent intensity by 8.1-fold (*p* < 0.001), 3.7-fold (*p* < 0.001), 4.2-fold (*p* < 0.001), and 1.3-fold (*p* < 0.01) compared to the fluorescent intensity observed in the HCD-consumed group, signifying substantial effect against HCD-induced ROS generation in the kidney. Among PCO1, PCO2, and PCO4, the most profound impact on inhibiting ROS production was displayed by PCO1, which was substantially (3.9-fold and 1.6-fold) better than the effect exerted by PCO2 and PCO4. The consumption of PCO3 was found ineffective in inhibiting HCD-induced ROS production.

Consistent with the ROS staining, the AO staining also displayed a higher extent of apoptosis in the HCD-consumed group, which was significantly 12.7-fold larger than the AO-stained area that appeared in the ND control group (Figure 6C,F). A significantly 8.4-fold (*p* < 0.001), 2.2-fold (*p* < 0.001) and 5.2-fold (*p* < 0.001) lower AO fluorescent intensity was observed in PCO1, PCO2 and PCO4 consumed groups compared to the HCD group, suggesting the effectiveness of PCO1, PCO2 and PCO4 to counter the HCD induced apoptosis. The consumption of PCO3 and PCO5 displayed no effect against HCD-induced apoptosis in the kidney section.

SA-β-gal staining revealed the higher cellular senescence in the HCD-consumed group, which was significantly (11.7-fold, *p* < 0.001) higher than the cellular senescence appearing in the ND control group (Figure 6D,G). The consumption of all the PCO (PCO1–PCO5) significantly (*p* < 0.001) countered HCD-induced cellular senescence. However, PCO1 and PCO2 consumption was most effectively apparent in the ~30-fold (*p* < 0.001) lower SA-β-gal positive cells compared to the HCD-consumed group. Among the PCOs, the worst effect on the cellular senescence was displayed by PCO3, which had a 14-fold, 14-fold, 5.4-fold, and 2.8-fold larger senescent area compared to the senescent area quantified in the case of PCO1, PCO2, PCO4, and PCO5, respectively. A combined result deciphered the effective role of PCO1, PCO2, and PCO4 in preventing HCD-induced adverse effects; precisely, PCO1 was the most effective in preventing kidney damage caused by HCD.

### 2.7. Examination of the Ovary Section

Figure 7A,E of the ovary section documented a high prevalence of previtellogenic (PV) oocytes in the HCD-consumed group. Compared to the ND-consumed group, 1.2-fold (*p* < 0.001) more PV oocytes were quantified in the HCD group. Also, the presence of degenerated PV oocytes (as indicated by the blue arrow) was detected in the HCD group. Compared to the HCD group, the PCO1-consumed group displayed a significantly (1.1-fold, *p* < 0.001) lower number of PV oocytes. Furthermore, the consumption of PCO2–PCO5 caused a non-significant effect on the PV oocyte counts compared to the HCD group.

Contrary to PV oocytes, significantly larger (5.0-fold, *p* < 0.01) mature vitellogenic (MV) oocyte counts were noticed in the PCO3-consumed group than in the HCD group, which had a similar MV oocyte count to that in the ND control group (Figure 7A,E). The PCO2-consumed group also displayed significantly (3.8-fold, *p* < 0.05) larger MV oocyte counts than the HCD group. Interestingly, non-significant changes in the early vitellogenic (EV) oocytes were observed between the groups. Results revealed a significant impact of PCO1 in preserving the ovary architecture disturbed by the consumption of HCD.

The consumption of HCD displayed a significantly (3.1-fold, *p* < 0.001) higher DHE fluorescent intensity than the ND control group, signifying the potential effect of HCD on the ROS generation in the ovary (Figure 7B,F). A significant 3.4-fold (*p* < 0.001), 1.6-fold (*p* < 0.05), 4.2-fold (*p* < 0.001), and 2.2-fold (*p* < 0.001) diminished ROS level was noticed in the PCO1, PCO2, PCO4, and PCO5 groups, respectively, documenting their impact against HCD-induced ROS production. On the contrary, no diminishing effect of PCO3 consumption was observed on HCD-induced ROS production in the ovary.

The AO fluorescent staining revealed a ~3-fold (*p* < 0.01) higher extent of apoptosis in the HCD-consumed group than in the ND-supplemented groups (Figure 7C,G). The consumption of PCO1, PCO2, and PCO4 significantly reduced HCD-induced apoptosis by 3.4-fold (*p* < 0.001), 1.8-fold (*p* < 0.01) and 2.0-fold (*p* < 0.01), respectively. On the contrary, PCO3 and PCO5 did not display a desired effect in countering HCD-induced apoptosis. Even more, PCO3 consumption aggravated the HCD-induced apoptosis in the ovary.

The ND group displayed the basal level of cellular senescence, which was significantly (18.6-fold, *p* < 0.001) increased by the HCD consumption (Figure 7D,H). The consumption of PCO1, PCO2, PCO4, and PCO5 had the significant effect of curtailing HCD-induced senescence, manifested by a 28.5-fold (*p* < 0.001), 5.9-fold (*p* < 0.01), 2.9-fold (*p* < 0.01), and 2.2-fold (*p* < 0.05) reduced SA-β-gal positive cell area compared to the HCD group. However, PCO1 displayed the most promising effect, as evidenced by the 4.8-fold and 10.2-fold reduced senescent stained area compared to PCO2 and PCO4, respectively. In contrast, a non-significant impact of PCO3 and PCO5 was observed against HCD-induced cellular senescence.

### 2.8. Examination of the Testis Section

The H&E staining of the testis section revealed a well-arranged, tightly packed tubular structure containing spermatocytes (ST) and spermatozoa (SZ) in the control group (Figure 8A,E). Contrary to the control group, the HCD group displayed a haphazard tubular structure with nebulous spermatocytes and spermatozoa with a significantly (1.8-fold, *p* < 0.001) higher interstitial space between the seminiferous tubules. PCO1, PCO2, PCO4, and PCO5 consumption substantially affected HCD-altered testicular morphology. In PCO1, PCO2, PCO4, and PCO5, a significant 2.4-fold (*p* < 0.001), 1.5-fold (*p* < 0.001), 1.3-fold (*p* < 0.001), and 1.2-fold (*p* < 0.001) reduced interstitial space was observed as compared to the HCD-consumed group. PCO3 consumption displayed no impact against HCD-induced testis damage, which was apparent by loosely arranged spermatocytes and spermatozoa with an extended interstitial area that was somewhat similar to that appearing in the HCD group.

The DHE fluorescent staining revealed a 3.6-fold (*p* < 0.001) higher DHE fluorescent intensity corresponding to the ROS level in the HCD group than in the ND control group (Figure 8B,F). PCO1 and PCO4 consumption efficiently curtailed HCD-induced ROS generation as documented by a significantly (~2-fold, *p* < 0.01) reduced DHE fluorescent intensity compared to that in the HCD group. No significant effect of PCO2, PCO3, and PCO5 consumption was noticed to reduce the ROS level actuated by the HCD consumption. Even more, PCO3 intensified the HCD-induced DHE fluorescent intensity, thus having a severe effect on the HCD-induced ROS generation.

The AO staining documented the heightened AO intensity representing apoptosis in the HCD-consuming group, which was further exacerbated by the consumption of PCO2 and PCO3 (Figure 8C,G). On the contrary, a significantly (3.7-fold, (*p* < 0.001) reduced AO fluorescent intensity was detected in the PCO1-consumed group compared to the HCD group, testifying to a substantial effect of PCO1 to curtail HCD-induced apoptosis in testes. 

A significantly (5.6-fold, *p* < 0.05) larger SA-β-gal positive cell area was observed in the HCD-consumed group compared to the ND group (Figure 8D,H). The HCD-induced cellular senescence was significantly (4.5-fold, *p* < 0.05) reduced by the consumption of PCO1 and PCO4 as compared to the HCD group. On the contrary, no impact of PCO2 and PCO3 was observed on the cellular senescence compared to the HCD group. Interestingly, PCO5 consumption logged a significantly (1.9-fold, *p* < 0.05) larger SA-β-gal positive cell area than the HCD group, suggesting the adverse effect of PCO5 against senescence inhibition in testes.

### 2.9. Multivariate Analysis

A multivariate analysis employing a principal component analysis (PCA) and hierarchical cluster analysis (HCA) was performed to differentiate the impact of the consumption of different PCOs against the HCD-induced stress based on the outcomes of zebrafish survivability, body weight change, dyslipidemia, inflammation and senescence in the liver, fatty liver changes, oxidative stress, apoptosis, and hepatic function biomarkers. The PCA scoring plot covered 82.9% of the variance and revealed the effect of HCD-PCO1 was close to the effect observed in the ND control group (Figure 9A). The HCD-PCO3 group was placed in the distal positive coordinates of principal component (PC) 1 and principal component (PC) 2 in the loading plot and represented a distinct effect compared to the impact exerted by the other groups. 

The HCA is considered a powerful tool for classifying different groups based on similarity. The HCA segregates all the groups into two major clusters (I and II) (Figure 9B). The HCD outcomes placed ND, HCD-PCO1, HCD-PCO2, and HCD-PCO4 in cluster I; however, a further analysis revealed the closest similarity to be between the ND and HCD-PCO1 groups, which was effectively different from that of the HCD-PCO2 and HCD-PCO4 groups. The HCD, HCD-PCO5, and HCD-PCO3 groups were placed in cluster II based on the similarity of their effects. Further, the HCD-PCO3 and HCD-PCO5 groups showed higher similarity, which differed from the HCD-consumed group. The multivariate outcomes confirmed that the different PCOs had distinct effects against HCD-induced changes in zebrafish, highlighting the functional superiority of PCO1 over other PCOs (PCO2–PCO5).

## 3. Discussion

In the present study, five distinct policosanol brands were examined for their comparative effect against HCD-induced alterations in zebrafish. The basic rationale behind the selection of the used brands was based on the variability in source material (policosanol) locations, which had been documented as a decisive factor for the diverse functionality of policosanols. In addition, the extra supplementation of LCAA (octacosanol) other than the native composition of the extracted policosanol (as in PCO5) and exclusive presence of octacosanol with different additives such as red yeast rice powder (as in PCO1) was the other important reasons for the selection of policosanol brands for the comparative study. Different policosanol brands (PCO1–PCO5) exhibited diverse effects while consumed by zebrafish during six weeks under the influence of an HCD. The difference in the activity emanated from the varying compositions and formulations of policosanols in the different brands. It has been well described that the policosanol’s composition and activity vary greatly based on its origin, source material, geographical location, and extraction method [14,27,28,29]. Like PCO1, PCO2 and PCO4 consist mainly of a mixture of distinct policosanols; however, the composition of different policosanols in PCO2 and PCO4 is not defined. On the contrary, the PCO1 is a well-defined mixture of eight different LCAAs [25,26] that work in a synergistic manner and are responsible for diverse biological activities. On the other hand, PCO5 contains octacosanol as a major ingredient of policosanol, while PCO3 exclusively has octacosanol (C28 of LCAA), along with a substantial amount of red yeast rice powder and thus has distinct biological activities as compared to PCO1, PCO2, and PCO4. The outcomes from the multivariate analysis of the present study established that octacosanol-containing brands’ (PCO2 and PCO3) impact on HCD-induced adverse effects in the zebrafish was distinctly inferior from the other policosanol-containing brands (PCO1, PCO4, and PCO5) and underscored the importance of a blend of long-chain aliphatic alcohols, as opposed to a single type of long-chain aliphatic alcohol, in shaping the biological activity.

The survivability of zebrafish after six weeks of consumption of different PCO brands remained almost similar to the survivability observed in the ND control group. However, a significant decline in zebrafish survivability was observed in the PCO3 group, signifying the toxicity of PCO3 towards zebrafish survivability. The HCD’s elevated body weight was curtailed by consuming different policosanols (PCO1–PCO5). The body weight reduction in the PCO groups is in accordance with the earlier findings documenting the impact of policosanols against obesity [14,30]. Moreover, in one of the clinical studies, Raydel-policosanol (PCO1) substantially reduced body weight in obese participants [18,19], signifying its functionality.

Additionally, the effect of policosanols on increasing the energy expenditure of adipose tissue has been observed as a key event in controlling obesity and obesity-associated disorders [14,31]. Also, some studies have documented the impact of policosanols on enhancing brown adipose tissue activity and consequently, a positive effect on diet-induced obesity [32]. Contrary to other PCOs, PCO3 consumption appeared with severe toxicity and abnormal body weight reduction. The apparent rationale for such events is the substantial amount of red yeast rice powder in PCO3 that is known to contain monacolin K [33,34]. This notion is supported by recent studies documenting a severe toxic effect of red yeast rice owing to the presence of monacolin K in it [35]. Even more recently, a Japanese company publicly apologized and withdrew a red yeast rice extract-based product containing monacolin K from the market due to the severe toxicity and mortality in consumers [36,37,38]. 

The impact of HCD consumption on hyperlipidemia is well established [39], and we also observed a severe adverse effect of HCD consumption on the blood lipid profile of the zebrafish. All the brands (PCO1–PCO5) displayed a substantial impact on reducing HCD-induced TG and TC levels and substantiated convincingly the earlier studies illustrating the effects of policosanol on balancing the lipid profile (TG and TC) [30,31]. The effect of policosanol on lipid profile maintenance is actuated by various events [14,40]. 

The principal mechanism behind the blood lipid-lowering role of policosanol is mediated by the phosphorylation of adenosine 5′ monophosphate-activated protein kinase (AMPK) that subsequently inhibits 3-hydroxy-3-methyl-glutaryl-coenzyme A (HMG-CoA) reductase, a rate-limiting enzyme in cholesterol biosynthesis leading to the alleviation of serum cholesterol [14,31]. Even more, hexacosanol (C26), an LCAA in policosanol, hinders the nuclear translocation of sterol regulatory element-binding protein (SREBP)-2, thus limiting cholesterol biosynthesis [40]. Moreover, the effect of policosanols as cholesteryl ester transfer protein (CETP) inhibitors [21] and on the HDL cholesterol efflux capacity [23] has been described as a key event behind their lipid-lowering role. Additionally, policosanols have been documented to induce cholesterol catabolism in the liver by inducing cholesterol conversion to bile acid and a subsequent fecal excretion leading to diminishing TC levels [24,41]. Moreover, policosanol has been documented to reduce apolipoprotein (apo-B) expression [24], thus inhibiting TG transfer and minimizing the blood TG level. Analogous to the mechanism of policosanol, the red yeast rice powder lowers cholesterol synthesis primarily by inhibiting HMG-CoA reductase due to the presence of monacolin K in it [11].

Consistent with the TC and the TG levels, all the brands (PCO1–PCO5) effectively alleviated the HCD-induced LDL-C levels. The basic reason behind the diminished LDL-C level in responses to PCOs is the inhibitory impact of policosanol on the apo-B expression and the enhancement of the LDL-C receptor in the liver that facilitates the uptake of the LDL-C from the blood [24]. However, the efficacy of reducing the LDL-C level differed between the brands (PCO1–PCO5). This was due to the different compositions and formulations of policosanols in different brands. The results are in accordance with the earlier reports documenting the varied effects of policosanols based on their source of origin and their composition [28,29]. Unexpectedly, the policosanol brands (PCO2–PCO5) were ineffective in boosting the diminished HDL-C levels caused by the consumption of HCD. However, PCO1 was found effective in elevating the HDL-C level, signifying its functional superiority over the other PCOs. The main reason for this activity is the distinct composition and ratio of policosanols in PCO1 [25]. The results align with the earlier reports suggesting the functional superiority of the policosanol composition in PCO1 compared to other policosanols originating from different sources and origins [28,29]. Even more, in a clinical study, PCO1 consumption for 2 weeks substantially enhanced the HDL-C level in Korean women compared to the placebo control [42], supporting the present findings. Additionally, PCO1 has been shown to inhibit CETP [21] and thus has a definite effect on enhancing HDL-C [43].

HCD is well known for its detrimental effect on the liver [44], which was effectively prevented by policosanol consumption. However, the tested policosanol brands exhibited a substantial variation in the hepatoprotective effect. The outcomes of the H&E staining suggested the non-effectiveness of PCO3 and PCO5 against HCD-induced hepatic damage. The presence of only a single type of long-chain aliphatic alcohol (octacosanol) in PCO3 and PCO5 is probably the reason for the limited hepatoprotective effect. On the contrary, PCO1, PCO2 and PCO4 have a mixture of high-molecular-weight aliphatic alcohols that work in a synergistic manner and display better hepatoprotective results. The hepatic histology results were consistent with the findings of plasma hepatic function biomarkers and indicate heightened AST and ALT levels in the PCO3 and PCO5 groups. Even more, the PCO3-consumed group displayed an augmentation in HCD-induced AST and ALT levels. We speculated the adverse effect of PCO3 was caused by the presence of red rice extract containing monacolin K, which has been documented to cause liver damage [45,46].

DHE fluorescent imaging revealed an elevated ROS level in the HCD-consumed group, which was significantly prevented by PCO1 and PCO4 consumption. However, no effect on HCD-induced ROS was detected in the PCO3, PCO4, and PCO5 groups, signifying the disparity between different policosanol brands owing to the composition of the policosanol (long-chain aliphatic alcohol) and its additives. However, policosanols are generally described as cellular antioxidants, mainly by triacontanol (C30 of LCAA) [14]. The presence of triacontanol (100–150 mg/g) in PCO1 [20] might be the leading cause of the diminished ROS level in this group. The results align with an earlier study documenting the cellular antioxidant nature of triacontanol, which prevents oxidative stress and lipid peroxidation [47]. Analogous to the DHE fluorescent staining, the AO staining revealed the effective role of PCO1 and PCO4 consumption against HCD-induced apoptosis. However, in the DHE staining, PCO3, PCO4, and PCO5 displayed no impact on the HCD-induced apoptosis, suggesting the impact of the policosanol formulation and source on their bio-functionality. The consumption of PCO1 and PCO4 coincided with the lowest levels of ROS, pivotal in averting apoptosis and aligned with the fact that oxidative stress triggered by ROS is a principal factor driving cellular apoptosis [48,49]. The higher apoptosis and ROS extent in PCO1 might be due to the presence of monacolin K in it, which has been described to induce cellular oxidative stress-mediated apoptosis via the regulation of MAPKs and the NFκB pathway [50]. HCD has been documented to elevate the hepatic inflammation mediated by the IL-6 production [51] that was effectively curtailed by the consumption of PCO1. However, the other PCO brands (PCO2–PCO5) were ineffective in inhibiting HCD-induced IL-6 levels. The distinct policosanol formulation in the tested brands is a major cause of their response against HCD-induced impairment. The notion is strongly advocated by earlier studies documenting different efficacies of distinct policosanols against hepatic apoptosis, inflammation, and ROS levels based on the composition and the source of the policosanol [51]. Additionally, PCO1, PCO2, and PCO4 consumption effectively suppressed HCD-induced cellular senescence, whereas PCO3 and PCO5 exhibited lower effectiveness. The notably higher ROS inhibitory capacity of PCO1, PCO2, and PCO4 primarily contributes to reducing cellular senescence, given that oxidative stress is widely acknowledged as a key factor triggering cellular senescence in various organs [52,53].

HCD-induced hyperlipidemia is one of the leading causes responsible for kidney [54] and reproductive organ impairment [55,56]. A substantial beneficial effect of policosanols on kidney function, inflammation, and apoptosis has been observed. Preceding studies documented the diverse impact of policosanols on the kidney, testes and ovaries owing to the composition of the long-chain aliphatic alcohols and source for the policosanol extraction [29,51]. Consistently, in the present study, we also noticed a distinct effect of different policosanol brands on kidneys, ovaries, and testes. Unlike PCO3 and PCO5, which mainly contain octacosanol, PCO1, PCO2, and PCO4 displayed better kidney protection by effectively reducing the HCD-induced ROS and apoptosis. The presence of different long-chain aliphatic alcohols in PCO1, PCO2, and PCO4 is the key contributor to these events that have been studied for their antiapoptotic role by modulating the activation of caspase 1 in the kidney [57]. The modulatory effect of policosanol on the high-mobility group box (HMGB)-1/phosphatidylinositol-3-kinase (PI3K)/mammalian target of rapamycin (mTOR)/nod-like receptor pyrin domain-containing-3 (NLRP3), and caspase-1 signaling pathway has been documented as the key mechanism in preventing kidney damage induced by HCD [57]. Among the different policosanol brands, PCO1 displayed the least effect against HCD-induced kidney damage. The presence of red rice extract in PCO1 is probably the reason for such an abnormal outcome. The notion is supported by a study depicting the acute kidney damage induced by the red yeast extract [58]. Studies on the policosanol effect on reproductive health are limited [29,59]. In one such study, the nontoxic nature of Cuban policosanol was documented in rats’ fertility and reproductive behavior [59]. Also, no evidence of external or visceral morphology alteration of the fetus was noticed, signifying policosanol’s nontoxic effect towards the embryonic development of rats [59]. Moreover, the comparative study between the policosanol originating from Cuba, China, and the USA displayed a variable impact on zebrafish ovary, testis morphology, and egg-laying behavior [29]. In the present study, different policosanol brands (PCO1–PCO5) displayed variations in their effect to prevent ovary and testis morphology from being altered by the consumption of HCD. The difference in activity between the PCOs is probably due to their source material origin and composition, which has been documented as the crucial element for the varied functionality of policosanol [14,29]. To further explore this study’s outcomes, future investigations will be carried out to examine the impact of different policosanol brands on reproductive health in terms of their impact on the egg-laying behavior, teratogenic effect, and survivability of the embryos. 

The basic limitation of the study is the short-term consumption of policosanol (six weeks); however, it was due to the high mortality perceived in the PCO3 group. To address this in future research, a lower dose of policosanol will be consumed for a long term (~24 weeks) to examine its effect on zebrafish. Furthermore, assessments will be conducted at various times after stopping policosanol consumption to determine the health implications following the cessation. The study will also explore the impact of distinct policosanol brands on reproductive health, assessing the fertility of zebrafish in terms of egg-laying behavior and the survivability of the laid eggs.

A comprehensive study concerning metabolomics and transcriptomics will be conducted to evaluate the modulatory effect of distinct policosanol brands on the metabolic pathway. The efficacy of policosanol brands will be compared with statins, focusing on their relative impact on blood lipid profile and hepatic health. In addition, a comprehensive study will be performed, including a wide range of commercially available policosanol brands originated from distinct raw materials like sugarcane wax, rice bran, wheat germ, maize, etc., to evaluate their comparative effect on blood lipid profile and functionality of vital organs of zebrafish. The results will help to establish a foundation for standardizing the composition of policosanol products to achieve optimum benefits.

## 4. Materials and Methods

### 4.1. Materials

Five distinct policosanol brands, viz., Raydel-policosanol (abbreviated code name: PCO1), Solgar-policosanol (abbreviated code name: PCO2), NutrioneLife-policosanol (abbreviated code name: PCO3), Mothernest-policosanol (abbreviated code name: PCO4), and Peter & John-policosanol (abbreviated code name: PCO5) were purchased from their respective websites. A detailed specification of the policosanol brands used is listed in Table 1 and Appendix A. All the other chemicals and reagents, unless otherwise stated, were of analytical grade and used as supplied. A detailed list of the used chemicals is provided in Appendix A.

### 4.2. Zebrafish Husbandry

Zebrafish (20 weeks aged, *n* = 196) were maintained at a 28 °C water temperature with controlled light and dark photoperiods of 14 h and 10 h, respectively, following the standard guidelines of the Animal Care and Committee and the Use of Raydel Research Institute (approval code RRI-20-003, date of approval 3 January 2020). Zebrafish (*n* = 28) were fed only with the normal tetrabit (ND, Tetrabit Gmbh D49304, Melle, Germany) while the other zebrafish (*n* = 168) were exclusively maintained on the high-cholesterol diet (HCD) for one month (Figure 10).

### 4.3. Preparation of High-Cholesterol Diet (HCD) and Policosanol-Supplemented HCD 

A high-cholesterol diet (HCD) was prepared by adding cholesterol (final 4%) in the normal tetrabit. For the preparation of policosanol-supplemented HCD, the tablets (from different brands) were individually ground to obtain the powder. The powdered tablets (PCO1–PCO5) were mixed independently with the HCD to obtain the five distinct policosanol brand-supplemented HCDs. The policosanol from each brand was blended to ensure uniformity, maintaining a consistent active policosanol content (final 1%, *wt*/*wt*) in each HCD. Before selecting a 1% amount, a dose–response curve between a 0.5% and 5% concentration range of different PCOs on the survivability of zebrafish during 3 weeks of consumption was determined (Appendix A). Based on that study, a 1% PCO concentration, which showed the optimum survivability, was selected for comparison.

### 4.4. Zebrafish Fed with Different Policosanol-Supplemented High-Cholesterol Diets (HCDs)

Zebrafish (*n* = 28) maintained on the ND constituted group I (control) and were further maintained on the ND for the additional six weeks. Zebrafish (*n* = 168) fed with the HCD were randomly allocated in six groups (*n* = 28/group) (Figure 10). The zebrafish in group II were fed with the HCD, while the zebrafish in groups III, IV, V, VI, and VII were fed the HCD supplemented with PCO1, PCO2, PCO3, PCO4, and PCO5, respectively. All the groups were maintained on their respective diet for the six weeks.

At different time points (on week 0, 2, 4 and 6), the body weight of the zebrafish from the different groups was evaluated. For the body weight analysis, zebrafish were anaesthetized by submerging them in 0.1% 2-phenoxyethanol solution for 2 min followed by immediate body weight measurement using an electronic weighing machine (Ohaus, Parsippany-Troy Hills, NJ, USA). Also, the survivability of the zebrafish across all the groups was examined during the six-week experimental period. 

### 4.5. Analysis of Plasma 

The blood of zebrafish from the distinct groups was collected from the heart just after scarifying zebrafish using hypothermic shock [60]; the blood was immediately collected in 3tubes containing 1 mM ethylenediaminetetraacetic acid (EDTA) in PBS and processed for centrifugation (3000 rpm, 10 min) to collect the plasma. The T-CHO and TGs, Cleantech TS-S (Walko Pure Chemical, Osaka, Japan) commercial kit was used to quantify total cholesterol (TC) and triglycerides (TG) levels in the blood, following the standard protocol suggested by the manufacturer. Similarly, the AM-202, AM102K, and AM103K commercial kits (Asan Pharmaceutical, Hwasung, Republic of Korea) were utilized to quantify high-density lipoprotein cholesterol (HDL-C), alanine transaminase (ALT), and aspartate transaminase (AST) levels, respectively. A detailed procedure for the plasma analysis is listed in Appendix A. The low-density lipoprotein cholesterol (LDL-C) was quantified using the Friedwald equation [TC–HDL-C–(TG/5)].

### 4.6. Histology and Immunohistochemistry (IHC)

After six weeks, zebrafish from each group were sacrificed by hypothermic shock [60], and the different organs (liver, kidneys, testes, and ovaries) were surgically removed and stored in 10% formalin. The organs were dehydrated using ethanol and fixed in paraffin wax to obtain the tissue section (7 μm thick). The different tissue sections (liver, kidney, testis, and ovary) were individually processed for the Hematoxylin and eosin (H&E) staining following a previously described method [61] to determine the morphological changes.

Oil red O (ORO) staining was performed following a previously described method [60]. In brief, the hepatic tissue section (7 μm thick) was covered with ORO stain following a 5 min incubation at 60 °C. Subsequently, the section was rinsed with 60% isopropanol followed by staining with Hematoxylin; after 30 s, the section was washed with water and visualized under a microscope.

The hepatic tissue was processed for IL-6 detection using IHC staining [62]. The hepatic section (7 μm thick) was pooled with the primary antibody (200× diluted, ab9324, Abcam, London, UK) specific against IL-6 and incubated overnight at 4 °C. Subsequently, the primary antibody was rinsed away, and the tissue section was exposed to 1000× diluted HRP-linked secondary antibodies, i.e., anti-IL-6 primary antibodies, and the section was developed using the chromogenic substrate EnVison + system-HRP polymer kit (Dako, Glostrup, Denmark).

### 4.7. Senescent-Associated β-Galactosidase Imaging

Detection of the senescent cells in the tissue (liver, kidney, ovary, and testis) was examined by senescent-associated β-galactosidase staining following an earlier-described method [63]. Initially, the tissue section (7 μm thick) was fixed with 4% paraformaldehyde. After a 5 min incubation at room temperature (RT), the section was rinsed twice with phosphate-buffered saline (PBS) and subsequently immersed in a 5-bromo-4-chloro-3-indolyl-β-D-galactopyranoside (X-gal) solution [prepared by dissolving 0.1 g X-gal in 100 mL of citrate-buffered saline (pH 5.9) comprising 5 mM each of ferric and ferrocyanide, 0.15 M NaCl, and 2 mM MgCl_2_]. After incubation in the dark for 16 hr, the stained tissue section was rinsed thoroughly with PBS and visualized under a microscope.

### 4.8. Fluorescence Imaging of the Tissue Section

Dihydroethidium (DHE) and acridine orange (AO) fluorescent staining were performed to detect the reactive oxygen species (ROS) and extent of apoptosis in the tissue section (liver, kidney, ovary, and testis) following a method described earlier [60]. In brief, 0.25 mL of 30 μM DHE solution was poured over the tissue section (7 μm thick) following a 30 min incubation in the dark at RT. Subsequently, the stained section was rinsed thoroughly with PBS and visualized under the fluorescent microscope at the excitation and emission wavelength of 585 nm and 615 nm, respectively. To perform the AO fluorescent staining, 0.25 mL of a 5 μg/mL AO solution was poured on the tissue section (7 μm thick). After a 30 min incubation in the dark at RT, the stained section was rinsed with PBS and visualized under the fluorescent microscope at the excitation and emission wavelengths of 505 nm and 535 nm, respectively.

### 4.9. Statistical Analysis

Statistical Package for the Social Science Software Program (SPSS, version 23.0, Inc., Chicago, IL, USA) was utilized to perform a one-way analysis of variance (ANOVA) followed by Dunnett’s post hoc analysis to establish the statistical difference between the groups at *p* < 0.05. The Minitab Statistical software version 21.4 was used for the multivariate analysis concerning the principal component analysis (PCA) and hierarchical cluster analysis (HCA). Three independent replicates were performed for each experiment, and the results are depicted as mean ± SEM.

## 5. Conclusions

Five brands (PCO1–PCO5) of policosanols displayed a distinct and diverse effect against HCD-induced hyperlipidemia, hepatic damage, oxidative stress, and cellular senescence in zebrafish. The policosanol brands (PCO1, PCO2, and PCO4) blended with distinct policosanols showed higher efficacy against the HCD-induced toxicity, contrary to the PCO brands (PCO3 and PCO5) containing octacosanol as the major ingredients. PCO1 (Raydel^®^), derived from Cuban sugarcane wax, exhibited the most significant effectiveness in improving HCD-induced mortality and hyperlipidemia and preventing damage to the liver, kidneys, testes, and ovaries. Conversely, PCO3 (containing red yeast rice), which exclusively contained octacosanol supplemented with red yeast extract, exhibited substantial exacerbation in HCD-induced mortality and dyslipidemia with elevated blood AST and ALT levels, indicating considerable hepatic toxicity. The findings uncovered a discrepancy in in vivo efficacy and unequal functionality among the policosanol brands.

## Figures and Tables

**Figure 1 pharmaceuticals-17-00714-f001:**
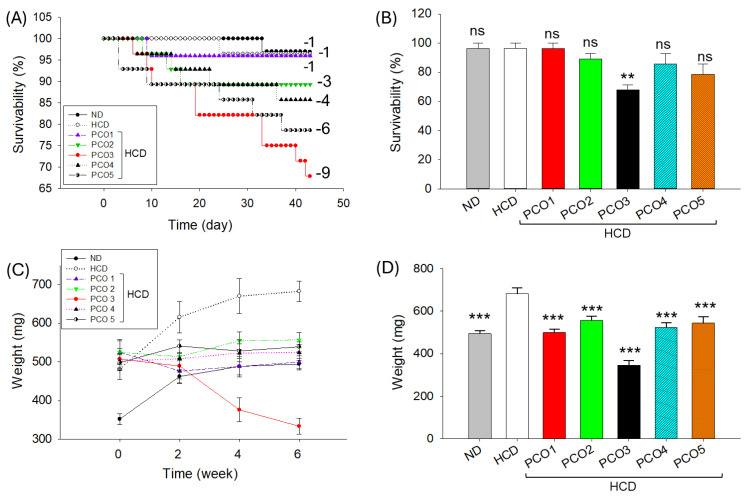
Effect of the consumption of different policosanols (PCO1–PCO5) under a high-cholesterol diet (HCD) for six weeks on the survivability and body weight of adult zebrafish. (**A**) Time-dependent [0–42 days] survivability of zebrafish. Numerical values (−1 to −9) represent each group’s dead zebrafish. (**B**) Zebrafish survivability at six weeks. (**C**) Time-dependent (0–6 weeks) changes in the body weight of zebrafish. (**D**) Zebrafish body weight at six weeks. Each value represents the mean value ± SEM of three individual investigations. ND represents the normal diet, HCD denotes the high-cholesterol diet [i.e., ND + cholesterol (final 4%, *wt*/*wt*), while HCD + PCO1/PCO2/PCO3/PCO4, or PCO5 represents the HCD supplemented with Raydel-policosanol, Solgar-policosanol, NutrioneLife-policosanol, Mothernest-policosanol, or Peter & John-policosanol (final 1%, *wt*/*wt*), respectively. The ** denotes *p* < 0.01, and *** denotes *p* < 0.001 compared to the HCD-consumed group, employing a one-way ANOVA followed by Dunnett’s post hoc analysis; ns denotes a non-significant difference between the groups.

**Figure 2 pharmaceuticals-17-00714-f002:**
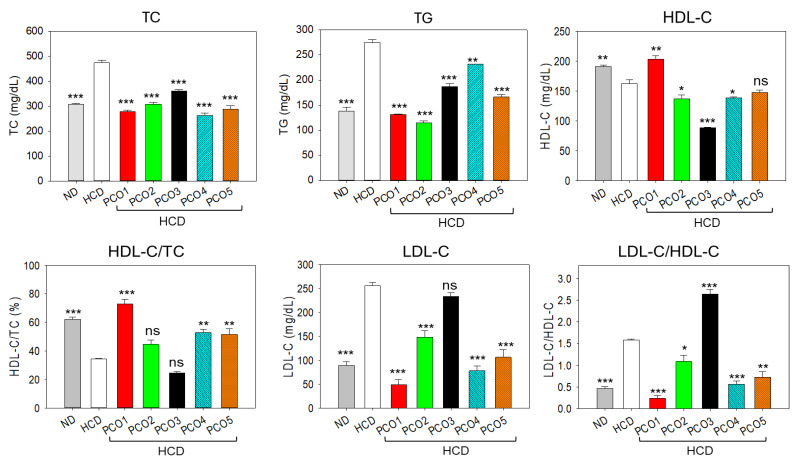
Effect of the consumption of different policosanols (PCO1–PCO5) under a high-cholesterol diet (HCD) for six weeks on the blood lipid profile of adult zebrafish. Each value represents the mean value ± SEM of three individual investigations. TC, TG, HDL-C, and LDL-C represent total cholesterol, triglycerides, high-density lipoprotein, and low-density lipoprotein cholesterol, respectively. ND represents the normal diet, HCD denotes the high-cholesterol diet, i.e., ND + cholesterol (final 4%, *wt*/*wt*), while HCD + PCO1/PCO2/PCO3/PCO4, or PCO5 represents the HCD supplemented with Raydel-policosanol, Solgar-policosanol, NutrioneLife-policosanol, Mothernest-policosanol, or Peter & John-policosanol (final 1%, *wt*/*wt*), respectively. The * denotes *p* < 0.05, ** denotes *p* < 0.01, and *** denotes *p* < 0.001 compared to the HCD-consumed group employing a one-way ANOVA followed by Dunnett’s post hoc analysis; ns denotes a non-significant difference between the groups.

**Figure 3 pharmaceuticals-17-00714-f003:**
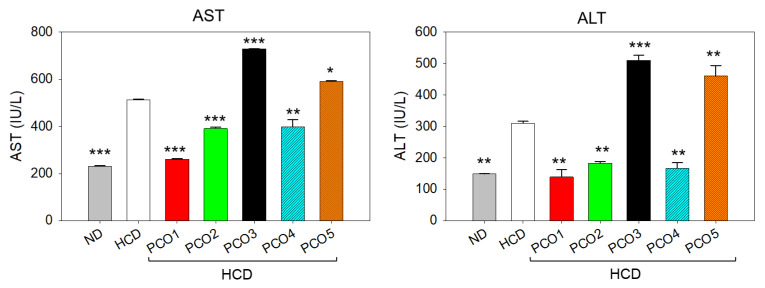
Hepatic function biomarkers aspartate aminotransferase (AST) and alanine transaminase (ALT) levels in the blood of adult zebrafish after six weeks of consumption of different policosanols (PCO1–PCO5) under a high-cholesterol diet (HCD). Each value in the bar graph represents the mean ± SEM of three individual investigations. ND represents the normal diet, HCD denotes the high-cholesterol diet, i.e., ND + cholesterol (final 4%, *wt*/*wt*), while HCD + PCO1/PCO2/PCO3/PCO4, or PCO5 represents the HCD-supplemented with Raydel-policosanol, Solgar-policosanol, NutrioneLife-policosanol, Mothernest-policosanol, or Peter & John-policosanol (final 1%, *wt*/*wt*), respectively. The * denotes *p* < 0.05, ** denotes *p* < 0.01, and *** denotes *p* < 0.001 compared to the HCD-consumed group employing a one-way ANOVA followed by Dunnett’s post hoc analysis.

**Figure 4 pharmaceuticals-17-00714-f004:**
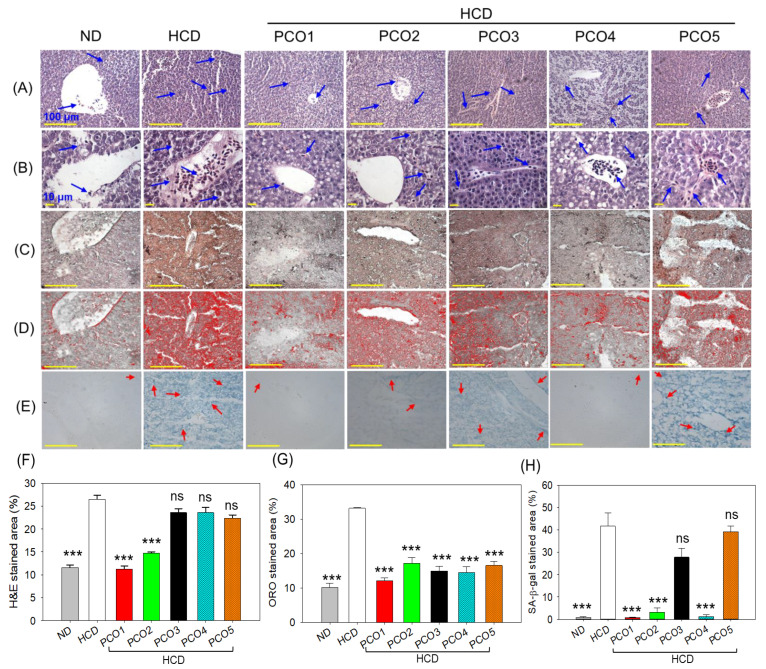
Effect of six weeks of consumption of different policosanols (PCO1–PCO5) under a high-cholesterol diet (HCD) on the liver of the adult zebrafish. (**A**,**B**) Images of the hematoxylin and eosin (H&E) stained area of the liver at 400× and 1000× magnifications. Blue arrows indicate neutrophil infiltration. (**C**) Oil red O (ORO) staining. (**D**) The ORO-stained area (brown color) was interchanged with red color using Image J software version 1.53 (http://rsb.info.nih.gov/ij/ accessed on 16 June 2023) to enhance the visibility [100 μm, scale bar]. (**E**) Senescent-associated β galactosidase (SA-β-gal) staining. Red arrows represent the senescent cells [100 μm, scale bar]. (**F**,**G**) Image J-based quantification of H&E and ORO-stained area; each value in the bar graph represents the mean ± SEM of six individual investigations. (**H**) Mean ± SEM value of SA-β-gal-stained area based on three individual investigations. ND represents the normal diet, HCD denotes the high-cholesterol diet [i.e., ND + cholesterol (final 4%, *wt*/*wt*), while HCD + PCO1/PCO2/PCO3/PCO4, or PCO5 represents the HCD supplemented with Raydel-policosanol, Solgar-policosanol, NutrioneLife-policosanol, Mothernest-policosanol, or Peter & John-policosanol (final 1%, *wt*/*wt*), respectively. *** denotes *p* < 0.001 compared to the HCD-consumed group employing a one-way ANOVA followed by Dunnett’s post hoc analysis; ns denotes a non-significant difference between the groups.

**Figure 5 pharmaceuticals-17-00714-f005:**
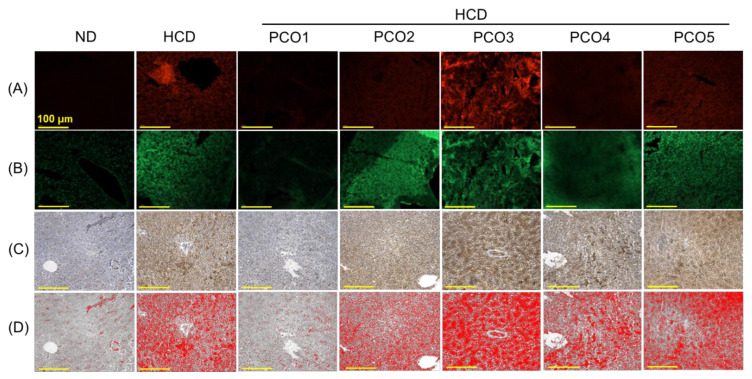
Effect of six weeks of consumption of different policosanols (PCO1–PCO5) under a high-cholesterol diet (HCD) on the reactive oxygen species (ROS), apoptosis, and interleukin (IL)-6 production on the liver of the adult zebrafish. (**A**) Dihydroethidium (DHE) fluorescent staining to examine the ROS level. (**B**) Acridine orange (AO) fluorescent staining to determine the extent of apoptosis. (**C**) Immunohistochemistry (IHC) for detecting IL-6 production. (**D**) The IL-6-stained area (brown color) was interchanged with red color using Image J software (at a brown color threshold value of 20–120) to enhance the visibility [100 μm, scale bar]. All the images are 400× magnified. (**E**,**F**) Quantification of fluorescent intensity corresponding to the DHE and AO-stained area. (**G**) Quantification of IL-6-stained area employing Image J software. Each value in the bar graph represents the mean ± SEM of six individual investigations. ND represents the normal diet, HCD denotes the high-cholesterol diet, i.e., ND + cholesterol (final 4%, *wt*/*wt*), while HCD + PCO1/PCO2/PCO3/PCO4, or PCO5 represents the HCD supplemented with Raydel-policosanol, Solgar-policosanol, NutrioneLife-policosanol, Mothernest-policosanol, or Peter & John-policosanol (final 1%, *wt*/*wt*), respectively. The * denotes *p* < 0.05, ** denotes *p* < 0.01, and *** denotes *p* < 0.001 compared to the HCD-consumed group employing a one-way ANOVA followed by Dunnett’s post hoc analysis; ns denotes a non-significant difference between the groups.

**Figure 6 pharmaceuticals-17-00714-f006:**
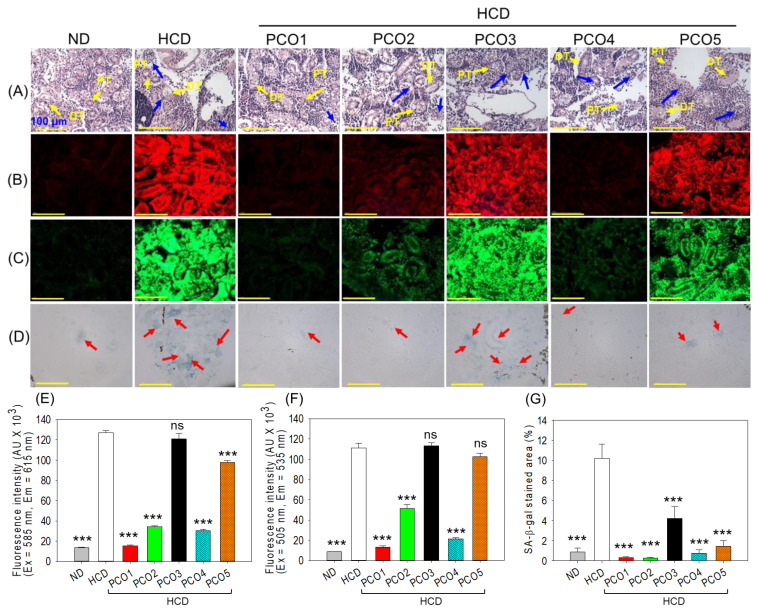
Effect of six weeks of consumption of different policosanols (PCO1–PCO5) under a high-cholesterol diet (HCD) on the kidneys of the adult zebrafish. (**A**) Hematoxylin and eosin (H&E) staining. PT and DT (represented by yellow arrow) are abbreviations for proximal and distal tubules, and the blue arrow depicts luminal derbies. (**B**,**C**) Dihydroethidium (DHE) and acridine orange (AO) fluorescent staining, respectively. (**D**) Senescent-associated β galactosidase (SA-β-gal) staining; the red arrow indicates the senescent stained cell. All the images are 400× magnified [Scale bar, 100 μm]. (**E**,**F**) are the Image J software-based quantifications of DHE and AO fluorescent intensity; each value in the bar graph represents the mean ± SEM of six individual investigations. (**G**) Mean ± SEM value of SA-β-gal-stained area based on three individual investigations. ND represents the normal diet, HCD denotes the high-cholesterol diet, i.e., ND + cholesterol (final 4%, *wt*/*wt*), while HCD + PCO1/PCO2/PCO3/PCO4, or PCO5 represents the HCD supplemented with Raydel-policosanol, Solgar-policosanol, NutrioneLife-policosanol, Mothernest-policosanol, or Peter & John-policosanol (final 1%, *wt*/*wt*), respectively. The *** denotes *p* < 0.001 compared to the HCD-consumed group employing a one-way ANOVA followed by Dunnett’s post hoc analysis; ns denotes a non-significant difference between the groups.

**Figure 7 pharmaceuticals-17-00714-f007:**
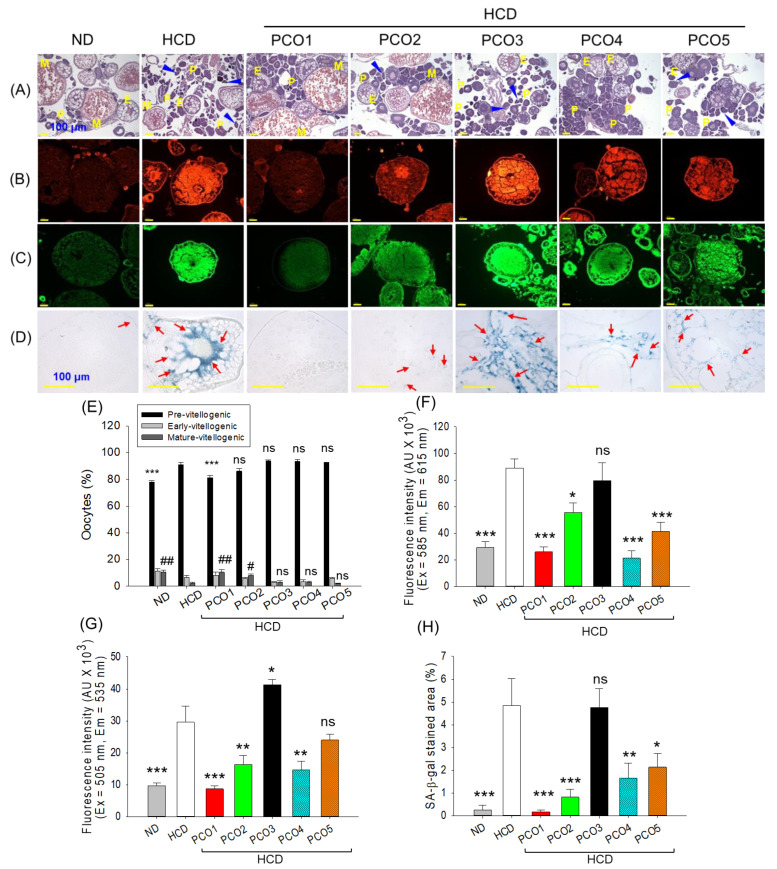
Effect of the intake of various policosanols (PCO1–PCO5) under a high-cholesterol diet (HCD) for six weeks on the ovary of the adult zebrafish. (**A**) Hematoxylin and eosin (H&E) staining. The P depicts premature, E depicts early, and M depicts the mature vitellogenic stage. The blue arrow points to a degenerated atretic previtellogenic follicle. Images are 200× magnified. (**B**) Dihydroethidium (DHE) and (**C**) acridine orange (AO) fluorescent staining. (**D**) Senescent-associated β galactosidase (SA-β-gal) staining; the red arrow indicates SA-β-gal positive cells, the senescent stained cells. Images are 400× magnified. (**E**) Quantification of different developmental populations of oocytes observed with H&E staining. (**F**–**H**) Quantification of DHE, AO fluorescent intensity, and SA-β-gal-stained areas, respectively, determined by Image J software. ND represents the normal diet, HCD denotes the high-cholesterol diet, i.e., ND + cholesterol (final 4%, *wt*/*wt*), while HCD + PCO1/PCO2/PCO3/PCO4, or PCO5 represents the HCD supplemented with Raydel-policosanol, Solgar-policosanol, NutrioneLife-policosanol, Mothernest-policosanol, or Peter & John-policosanol (final 1%, *wt*/*wt*), respectively. The * denotes *p* < 0.05, ** denotes *p* < 0.01, and *** denotes *p* < 0.001 compared to the HCD-consumed group, while ^#^ denotes *p* < 0.05, and ^##^ denotes *p* < 0.01 compared to the HCD-consumed group for mature vitellogenic oocytes employing a one-way ANOVA followed by Dunnett’s post hoc analysis; ns denotes a non-significant difference between the groups.

**Figure 8 pharmaceuticals-17-00714-f008:**
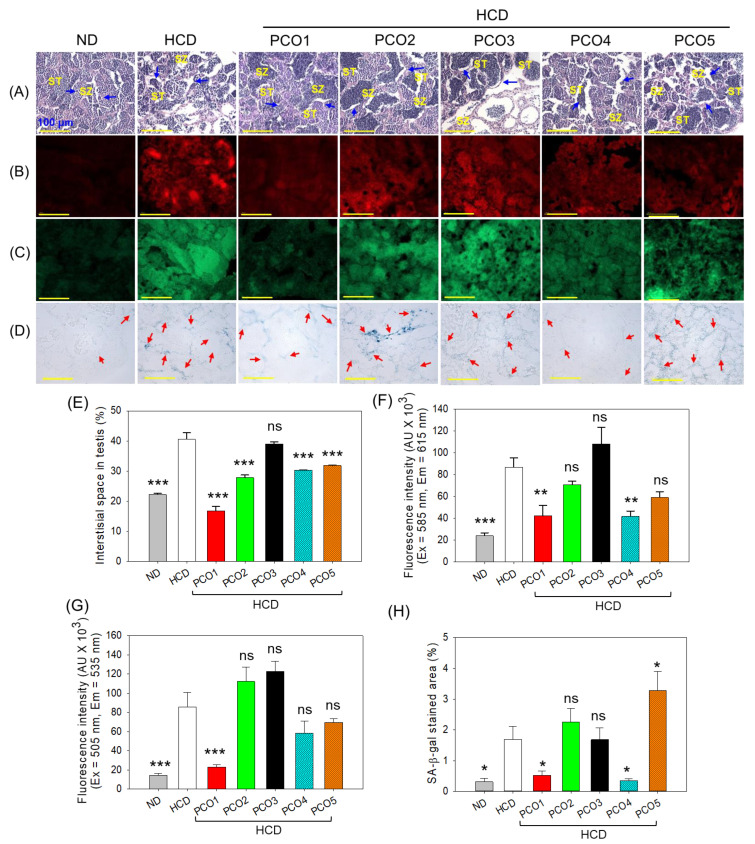
Effect of the consumption of different policosanols (PCO1–PCO5) with a high-cholesterol diet (HCD) for six weeks on adult zebrafish testes. (**A**) Hematoxylin and eosin (H&E) staining. ST depicts spermatocytes, and SZ depicts spermatozoa; the blue arrow highlights the interstitial space amid the seminiferous tubules. (**B**) Dihydroethidium (DHE) and (**C**) acridine orange (AO) fluorescent staining. (**D**) Senescent-associated β-galactosidase (SA-β-gal) staining; the red arrow highlights the senescent stained cell. All the images are 400× magnified [Scale bar, 100 μm]. (**E**) Quantification of interstitial space in the testis section. (**F**–**H**) Assessment of DHE, AO fluorescent intensity, and SA-β-gal-stained area by Image J software, respectively. ND represents the normal diet, HCD denotes the high-cholesterol diet, i.e., ND + cholesterol (final 4%, *wt*/*wt*), while HCD + PCO1/PCO2/PCO3/PCO4, or PCO5 represents the HCD supplemented with Raydel-policosanol, Solgar-policosanol, NutrioneLife-policosanol, Mothernest-policosanol, or Peter & John-policosanol (final 1%, *wt*/*wt*), respectively. The * denotes *p* < 0.05, ** denotes *p* < 0.01, and *** denotes *p* < 0.001 compared to the HCD-consumed group employing a one-way ANOVA followed by Dunnett’s post hoc analysis; ns denotes a non-significant difference between the groups.

**Figure 9 pharmaceuticals-17-00714-f009:**
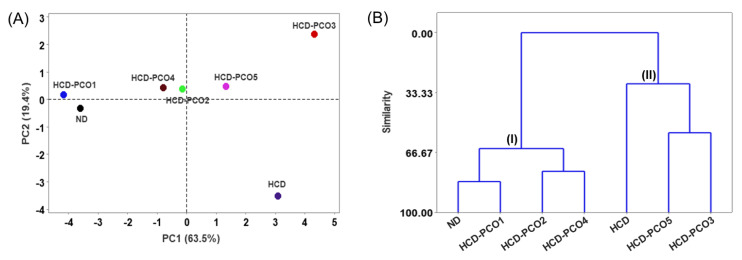
Multivariate analysis of the outcomes (survivability, body weight, blood lipid profile, hepatic function biomarkers, and hepatic histopathological analysis) after six weeks of consumption of different policosanols (PCO1–PCO5) with a high-cholesterol diet (HCD) by adult zebrafish. (**A**) Principal component analysis-derived loading plot. (**B**) Hierarchical cluster analysis (HCA). The multivariate analysis was performed by utilizing the Minitab Statistical Analysis software (version 21.4). ND represents the normal diet, HCD denotes the high-cholesterol diet, i.e., ND + cholesterol (final 4%, *wt*/*wt*), while HCD-PCO1/PCO2/PCO3/PCO4, or PCO5 represents the HCD supplemented with Raydel-policosanol, Solgar-policosanol, NutrioneLife-policosanol, Mothernest-policosanol, or Peter & John-policosanol (final 1%, *wt*/*wt*), respectively.

**Figure 10 pharmaceuticals-17-00714-f010:**
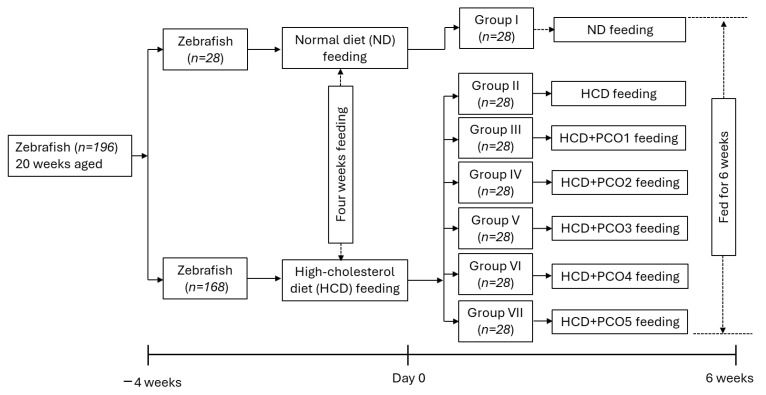
A schematic representation of the experimental design. ND represents normal diet, HCD denotes the high-cholesterol diet, i.e., ND+ cholesterol (final 4%, *wt*/*wt*), while HCD+ PCO1/PCO2/PCO3/PCO4, or PCO5 represents the HCD supplemented with Raydel-policosanol, Solgar-policosanol, NutrioneLife-policosanol, Mothernest-policosanol, or Peter & John-policosanol (final 1%, *wt*/*wt*), respectively.

**Table 1 pharmaceuticals-17-00714-t001:** A comparative ingredients/raw material in the five commercially available policosanol brands.

Product Code	Product Manufacturer Name and Country	Country of Origin (Source Material)	Source Material	Major Dose	Additive
PCO1	Raydel-policosanol, Thornleigh, NSW, Australia	Cuba	Sugarcane Wax	Policosanol 20 mg	None
PCO2	Solgar-policosanol, Leonia, NJ, USA	USA	ND	Policosanol 20 mg	None
PCO3	NutrioneLife-monacosanol, Seoul, South Korea	USA and India	ND	Octacosanol ^(1)^ 12 mg	RYR ^(2)^ 5 mg
PCO4	Mothernest-policosanol, Seven Hills, NSW, Australia	Australia	Sugarcane	Wax alcohol 20 mg	None
PCO5	Peter & John-policosanol, Auckland, New Zealand	New Zealand	ND	Policosanol 33.4 mg (octacosanol 20 mg)	None

ND: not disclosed; RYR: red yeast rice (powder); ^(1)^ raw material sourced from the USA; ^(2)^ raw material sourced from India.

## Data Availability

The data used to support the findings of this study are available from the corresponding author upon reasonable request.

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
