# Peer review of "Efficacy Assessment of Five Policosanol Brands and Damage to Vital Organs in Hyperlipidemic Zebrafish by Six-Week Supplementation: Highlighting the Toxicity of Red Yeast Rice and Safety of Cuban Policosanol (Raydel®)"

_pharmaceuticals, 2024, doi:10.3390/ph17060714_

Round 1
Reviewer 1 Report
Comments and Suggestions for Authors In the current manuscript, Cho and colleagues investigated the complex bioactivity of several commercially available policosanols with a focus on high-fat diet (HFD)-induced disorders. The authors conducted a screening of five policosanols in the HFD-fed zebrafish model, analyzing their effects on animal viability and weight growth, levels of key lipid markers in blood, such as cholesterol, triglycerides, low-density and high-density lipoprotein cholesterol, key liver markers ALT and AST, and protective effects on liver, kidney, ovary and testis (policosanol effects on histological changes, ROS and IL6 production, apoptosis induction and cell senescence). As a result of the study, Cho et al. revealed that policasonol 1, called Raydel, showed the most pronounced protective influence on all mentioned parameters compared to other substances. In my opinion, this manuscript can be published in Pharmaceuticals, but after a major revision. 1. The main problem of this paper is related to the obvious conflict of interest of the authors. It should be realized that the authors of this paper are employees of the Rydel Research Institute working for the Korean Medical Venture Center (see https://raydellab.modoo.at/), and the identified most bioactive substance is PC1 (also known as Raydel, the substance promoted by this venture center), which may contribute to the objectivity of this study. Of greater concern is the critically small sample size used by the authors to demonstrate the bioactivity of Raydel (only 3 out of 28 samples). Given that there were 28 animals each in the control and experimental groups, the use of 3 samples/group for statistical analysis is highly questionable. With such a small number of samples, there is a high probability of subjective error in animal tissue studies. In my opinion, to obtain more reliable results, the authors should increase the number of samples for analysis to at least 6 (for histological materials). In the absence of these data, the realism of the present work is questionable. 2. Table 1 - Please complete the table with information on the complete chemical composition of the listed policonasol provided by the manufacturers. 3. Figure 1B - Is the reported p-value (*) a log-rank p-value? If not, please correct. Please reanalyze the results - it is possible that the PCO5 group is also statistically different from the HCD group. 4. Figure 1C - Please use the same colors to draw the graphs as in Figure 1A to better guide the reader. 5. Lines 143-144 - due to the lack of statistical analysis data between groups PCO1, PCO4 and PCO5, the phrase that PCO1 shows the most striking results is not true. Based on the SDs shown in Figure 1D, the weights of mice from PCO1, PCO4, and PCO5 groups are very comparable. Please double check, please provide p-values. 6. Figure 2 - Please recalculate the p-value for PCO3 group - based on the above graph and the SD spread in the HCD and PCO3 groups, the differences are still statistically significant. The same applies to the bar graph for HDL-C (HCD vs. PCO4).Author Response
Thank you for your valuable comments and suggestions.
Please find attached doc as point-to-point response.

Reviewer 2 Report
Comments and Suggestions for Authors
1. Expand on the discussion regarding the biochemical pathways and mechanisms by which policosanols and red yeast rice might affect lipid metabolism, hepatic function, and overall zebrafish health.
2. Consider a follow-up study or add a section discussing potential long-term effects of policosanol supplementation, including recovery or further changes in zebrafish health after cessation of supplementation.
3. Provide a rationale for the selection of each policosanol brand and discuss any potential variability in source materials that might affect the study outcomes. Consider standardizing the source material for future studies.
4. Include a discussion on the potential reproductive health implications of policosanol supplementation, supported by relevant literature. Consider adding fertility assays or long-term reproductive health assessments in future studies.
5. Conduct additional experiments with varying doses of each policosanol brand to establish a dose-response relationship. This will help identify the optimal dose that maximizes efficacy while minimizing toxicity.
6. Add metabolomic studies or gene expression analyses to elucidate the specific metabolic pathways impacted by the different policosanol brands. This could provide deeper insights into the mechanisms of action and potential side effects.
7. Add a control group treated with a commonly used statin to directly compare the efficacy and safety profiles of policosanol and statins. This comparison would provide valuable context for the potential clinical use of policosanol as an alternative or complementary therapy to statins.
Author Response
Thank you for your valuable comments and suggestions.
Please find attached doc as point-to-point response.

Round 2
Reviewer 1 Report
Comments and Suggestions for Authors
The authors successfully responded to all of my comments, including the key question about the small sample sizes used for analysis. The authors doubled the number of samples analyzed, which greatly improved the objectivity of this study. The manuscript is now ready for publication. I express my deepest respect to Prof. Kyung-Hyun Cho and his team and wish them further success in their research.
Reviewer 2 Report
Comments and Suggestions for Authors
I have no comments